# Can Faecal Zonulin and Calprotectin Levels Be Used in the Diagnosis and Follow-Up in Infants with Milk Protein-Induced Allergic Proctocolitis?

**DOI:** 10.3390/nu16172949

**Published:** 2024-09-02

**Authors:** Grażyna Czaja-Bulsa, Karolina Bulsa, Monika Łokieć, Arleta Drozd

**Affiliations:** 1Chair and Department of Paediatrics and Paediatric Nursing, Pomeranian Medical University, 70-204 Szczecin, Poland; 2Szczecin Outpatient Clinic, 71-050 Szczecin, Poland; k.bulsa@gmail.com; 3Clinical Department of Paediatrics University Hospital, 65-046 Zielona Góra, Poland; monika.taczalska@op.pl; 4Department of Human Nutrition and Metabolomics, Pomeranian Medical University, 70-204 Szczecin, Poland; arleta.drozd@pum.edu.pl

**Keywords:** cow’s milk allergy, non-IgE-mediated allergy, food-protein-induced allergic proctocolitis, FPIAP, calprotectin, zonulin, children

## Abstract

Objective: The aim of our study was to investigate whether a 1-month-long milk-free diet results in a reduction in faecal calprotectin (FC) and faecal-zonulin-related proteins (FZRP) in children with milk-protein-induced allergic proctocolitis (MPIAP). Materials and methods: This is a single-centre, prospective, observational cohort study involving 86 infants with MPIAP, aged 1–3 months, and 30 healthy controls of the same age. The FC and FZRP were marked using the ELISA method (IDK^®^ Calprotectin or Zonulin ELISA Kit, Immunodiagnostik AG, Bensheim, Germany). The diagnosis of MPIAP was confirmed with an open milk challenge test. Results: FFC and FZRP proved useful in evaluating MPIAP treatment with a milk-free diet, and the resolution of allergic symptoms and a significant (*p* = 0.0000) decrease in the concentrations of both biomarkers were observed after 4 weeks on the diet. The FC and FZRP concentrations were still higher than in the control group. A high variability of FC concentrations was found in all the study groups. An important limitation is the phenomenon of FZRP not being produced in all individuals, affecting one in five infants. Conclusions: FC and FZRP can be used to monitor the resolution of colitis in infants with MPIAP treated with a milk-free diet, indicating a slower resolution of allergic inflammation than of allergic symptoms. The diagnosis of MPIAP on the basis of FC concentrations is subject to considerable error, due to the high individual variability of this indicator. FZRP is a better parameter, but this needs further research, as these are the first determinations in infants with MPIAP.

## 1. Introduction 

Cow’s milk protein allergy (CMPA) is the most common food allergy in infants and children younger than 3 years of age [1,2]. Its prevalence in developed countries is estimated at 0.5–3% in the first year of life [3,4,5]. If the gastrointestinal tract is involved, it is usually a non-IgE mediated allergy (non-IgE). One of its forms is milk-protein-induced allergic proctocolitis (MPIAP). The intestinal forms of non-IgE-CMPA cause intestinal mucositis, which is responsible for numerous symptoms. 

MIAP is the most common form of FPIAP (food-protein-induced allergic proctocolitis) [6]. The population prevalence in infants is approximately 1–2%, and the proportion of infants with rectal bleeding is 18–64% [7]. Affected infants are in generally good condition. Stools containing blood or blood strands usually appear during breastfeeding at 4–6 weeks of age; in some infants they can disappear spontaneously; even in a few weeks. Therefore, some researchers recommend observing the infant for 2–4 weeks without including milk-free diets [8,9]. Milk-free diets usually lead to the resolution of symptoms within 3–5 days, although in some people, the process takes several weeks [10,11,12,13]. Recent studies indicate that the majority (60%) of children with FPIAP develop tolerance to milk at around 12 months of age, although symptoms may persist up to 3 years of age [11,14]. A milk-free diet is usually maintained until the age of 12 months, when a follow-up OFC is performed to check whether milk tolerance has already developed [14]. In infants with FPIAP who underwent endoscopy, an equivocal picture was found. FPIAP is usually a patchy disease [15]. Histopathological findings are not pathognomonic of the disease [16]. Some infants have iron-deficiency anemia, and less frequent hypoalbuminemia and peripheral eosinophilia [17].

The only way to confirm the diagnosis of this type of allergy is to perform an elimination and milk challenge test, i.e., to eliminate milk from the diet, which should lead to the disappearance of symptoms, which recur when milk is reintroduced into the diet [18]. 

In the absence of simple diagnostic tests to confirm non-IgE CMPA, there has been increasing interest over the years in various faecal biomarkers used to diagnose and control the course of gastrointestinal allergic diseases. The first to be investigated was faecal calprotectin (FC), which is an indicator of intestinal mucositis in other diseases. The first study, conducted by Waligura-Dupriet at al. in 2011, reported that the concentrations of FC in infants with food allergies were two-fold higher [19]. Subsequent studies have confirmed significant variability in FC concentrations in healthy infants, as well as in infants with CMPA, indicating the need for further research [20]. 

Faecal-zonulin-related proteins (FZRPs) may be new non-invasive markers for the presence of food allergies. There is a family of structurally and functionally related proteins called zonulin-related proteins (ZRP) [21]. Higher FZRP activity can increase the passage of antigens through the paracellular pathway of the intestinal epithelium, which can lead to the abolition of immune tolerance, i.e., the onset of a food allergy [10]. Through tight intercellular junctions, the intestinal epithelial barrier, together with the intestinal lymphoid tissue and neuroendocrine network, can control the balance between tolerance and allergy to antigens other than one’s own. FZRPs are the physiological modulators of intercellular tight junctions. Their activity can significantly affect the occurrence of either food tolerance or allergies [22,23]. FZPR secretion has been shown to be regulated by pro-inflammatory cytokines, mainly IL-6 [24]. These scientific data suggest that FZRP may be used as a marker for diseases associated with inflammation in the gut and the development of food allergies. 

The aim of our study was to investigate whether a 1-month-long milk-free diet in infants with MPIAP results in a reduction in faecal FC and FZRP.

## 2. Materials and Methods 

This is a single-center prospective observational cohort study of 86 children with MPIAP, with a median age of 2 months (1–3 months), and 30 healthy controls, who were examined for faecal CF and FZRP. The study and control groups did not differ when it came to age, sex, weight and height, family history of allergic diseases, and feeding method at the time of diagnosis. They differed only in the prevalence of general symptoms, such as anxiety and anaemia, which were more common in children with MPIAP. Detailed characteristics of the study and control groups are given in Table 1.

This prospective study was conducted for 2 years (2020–2022). The infants were patients at the Outpatient Clinic for Paediatric Gastroenterology and Allergology in Szczecin. The patients lived in the West Pomeranian region and were selected from among children with symptoms indicative of CMPA—diarrhoea with blood or blood strands. Some infants also had atopic dermatitis.

All infants examined were in good general condition, with normal weight and height (10c–97c). The diagnostic diagram is shown in Figure 1.

In 98 infants with complete or significant improvement during treatment with a milk-free diet, an open oral food challenge test (OFC) with milk was performed. A positive OFC was the basis for the diagnosis of CMPA. A negative provocation test ruled out the presence of CMPA. All infants developed symptoms 8 h to 3 days after milk consumption. Infants with delayed reactions during OFC were diagnosed with non-IgE CMPA, and with MPIAP due to presenting symptoms. OFC was positive in 86 infants who were included in the study group (MPIAP_0_ group). The first stool sample for the determination of FC and FZRP was taken immediately before the introduction of the milk-free diet (FC_0_, FZRP_0_), and the second was taken after 4 weeks of the diet (FC_1_, FZRP_1_) (Figure 1).

During the diagnostic elimination diet and subsequent treatment of MPIAP, a milk-free diet was used: either milk from mothers remaining on a milk-free diet, extensively hydrolysed cow’s milk formula (eHF), or mixed feeding (breast and eHF). No infants were on an elementary diet (AAF—free amino acids formulae) (Table 1).

Each patient had a medical and allergological history taken (in which recurrent adverse reactions were recorded) and went through physical examination (current MPIAP symptoms). The criteria for including a child in the study were the presence of MPIAP, cow’s-milk-only-dependent symptoms, age (up to 4 months), no coexisting chronic diseases (except atopic dermatitis, as a form of CMPA), and written consent from the parents/legal guardians for the child to participate in the controlled study. Consent also included permission to store and publish the collected data. Exclusion criteria were as follows: another form of non-IgE-CMPA, coexistence of MPIAP with other chronic diseases, age above 4 months, and lack of written consent from parents/guardians for the child’s participation in the controlled study.

Children in the study group at the time of MPIAP diagnosis were designated as group MPIAP_0_, and assessed after one month of treatment with a milk-free diet as group MPIAP_1_.

The age of MPIAP diagnosis was the age when a diagnostic milk-free diet was introduced. The diagnosis of MPIAP (FPIAP) was given according to the recommendations of the WAO and EAACI [25,26].

The research was approved by the PUM Bioethics Committee, No. KB-0012/5/20. The research was financed by statutory activities (WNoZ-319-01/s/12/2020-2022). The presented results are part of the ongoing project.

### 2.1. OFC Procedures

OFC procedures were always initiated in an outpatient setting, under the supervision of a nurse and/or doctor, with access to anti-shock medication. These were provocations performed with the usage of an open method [27]. After a negative lip test (a drop of milk), gradually, higher doses of milk were administered every 15 min (1 mL, 2 mL, 5 mL, 10 mL, 20 mL, 50 mL, and up to 100 mL). Patients remained under observation for at least 2 h (usually 4–6 h) from the end of OFC [27,28,29] The provocation was continued at home for another 6 days. Every day, parents administered the milk mixture corresponding in volume to one meal (up to 120 mL), and the information about possible adverse reactions was recorded in the observation card. After 6 days (or earlier, if side effects had occurred), the doctor examined the OFC outcome. In the study, results of 86 milk OFCs were analysed.

### 2.2. Faecal Samples

The first stool sample was taken immediately before the introduction of the milk-free diet (FC_0_, FZRP_0_), and the other after 4 weeks of the diet, when the symptoms had disappeared or decreased (FC_1_, FZRP_1_) (Figure 1). Raw stool samples from all children were frozen and stored at −80 °C. All patients provided a stool specimen the day before their visit to the outpatient clinic.

The FC was determined using the ELISA method (IDK^®^ Calprotectin ELISA Kit, Immunodiagnostik AG, 64625 Bensheim, Germany). The FC results were given in µg/mL. According to the manufacturer, the normal range for FC was set at <50 µg/g for adults and children over 4 years of age. Values of 50–100 µg/g are regarded as borderline and >100 µg/g as positive. The company did not specify a reference range for younger children [8]. 

The FZRP was also assessed with the use of the ELISA method (IDK^®^ Zonulin ELISA Kit, Immunodiagnostik AG, Germany). The manufacturer indicates that the correct median concentration of FFRZ is 61 ng/mL. The manufacturer states that the intra-assay and inter-assay coefficients of variation were 3.4%, and 13.3%, respectively.

Olafsdottir et al. showed that stool collected from the nappy had a 30% higher FC concentration due to water absorption [18]. In all infants, stools were collected in the same way (from the nappy) for the determination of FC and FZRP.

### 2.3. Statistical Analysis 

All data were collected in electronic form in MS Excel spreadsheet and were subjected to statistical analysis. Continuous variables were described by median, minimum, and maximum values. Discontinuous variables were described by number and frequency of occurrence. Pearson’s χ^2^ test or Fisher’s exact test and Spearman’s rank correlation were used to test statistical relationships between discontinuous variables.

Probability p was calculated with two tests: Mann–Whitney U-test for variables, for which the normality of the distributions was not satisfied, and the Student’s *t*-test for variables with a normal distribution.

In the search for the FC concentration that differentiates healthy children (control group) from children diagnosed with MPIAP best (MPIAP_0_ group), ROC curve analysis was used. The results were described by the area under the curve (AUC), the standard error of the AUC (SE), the 95% confidence interval for the AUC (95% CI), the p-likelihood, and the coordinates of the ROC curves, i.e., the sensitivity and specificity of the study group relative to the control group were estimated for each range of values of the continuous variable.

## 3. Results 

### 3.1. Faecal Calprotectin (FC)

The median FC concentration in children in the control group was 113.25 mg/L (13.9–219.9 mg/L) (Table 2). It was significantly lower (*p* = 0.0000) than in the children in the study group at the time of MPIAP diagnosis (MPIAP_0_), 382.9 mg/L (103.5–822.8 mg/L), and in the children in the study group after 4 weeks of treatment with a milk-free diet (MPIAP_1_) 208.4 mg/L (67.9–484.4 mg/L) (*p* < 0.0001). At this time, the children’s CMPA complaints (MPIAP and atopic dermatitis) had resolved. On a milk-free diet, the FC concentrations decreased significantly in each infant (Figure 1). 

The FC concentrations differed significantly between the children in the MPIAP_1_ and MPIAP_0_ groups (*p* < 0.0001) and were correlated positively (r = 0.76, *p* < 0.0001). High variability in the faecal FC concentration was found in all the study groups (control group, and MPIAP_0_ and MPIAP_1_ groups) (Figure 1).

In the search for the FC concentration that differentiated the healthy children (control group) from the children diagnosed with MPIAP best (MPIAP_0_ group), a ROC curve analysis was used. It was found that the FC concentration > 193.75 mg/L for the MPIAP_0_ group (with a sensitivity of 92% and specificity of 89%) distinguished it from the infants from the control group in the best way (FC < 193.75 mg/L). Lower values were found in 96.7% of the children in the control group, with values equal or higher in 91.9% of the children in the MPIAP_0_ group and 61.6% of the children in the MPIAP_1_ group (Table 3).

In all the study groups, the level of FC did not depend on gender, family history of allergy, or the type of feeding (breast, modified milk, mixed diet).

### 3.2. Faecal-Zonulin-Related Proteins (FZRP)

The median FZRP concentration in the children in the control group was 54.1 ng/mL, and the range was 36.6–101.9 ng/mL (Table 2). It was significantly lower (*p* = 0.000) than in the children in the study group at the time of MPIAP diagnosis (MPAIP_o_), 103.6 ng/mL (67.1–378.7 ng/mL), and in the children in the study group after 4 weeks of treatment with a milk-free diet (MPIAP_1_) 62.9 ng/mL (13.3–143.3 ng/mL) (*p* < 0.0001). By this time, intestinal complaints and skin lesions had resolved in the children. On a milk-free diet, the FZRP concentrations decreased significantly in each infant (Figure 2). The FZRP concentrations differed significantly between the children in the MPIAP_1_ and MPIAP_0_ groups (*p* < 0.0001) and were correlated positively (r = 0.72, *p* < 0.0001).

In all the study groups, the FZRP concentrations did not depend on gender, type of feeding (breast, modified milk), or family history of allergic diseases.

In the control group, the FZRP concentrations were correlated positively with the FC concentrations (r = 0.07, *p* = 0.0026). This relationship was not observed in the study groups (MPIAP_0_, MPIAP_1_).

In the search for the FZRP concentration that best differentiated healthy children from the control group with children diagnosed with MPIAP, a ROC curve analysis was used. It was found that an FZRP concentration ≥66.28 ng/mL for the MPIAP_0_ group (with sensitivity of 100% and specificity of 83%) differentiated it from the control group the best (FZRP < 66.28 ng/mL). Values lower than 66.28 ng/mL were found in 83.3% of the children in the control group and values equal or higher in 100.0% of the children in the MPIAP_0_ group and 45.6%of the children in the MPIAP_1_ group (Table 3).

Zonulin was not produced by 16 children (20.6%), all of whom were in the MPIAP group (18.6%).

## 4. Discussion

FC and FZRP are useful for monitoring the resolution of colitis in infants with MPIAP treated with a dairy-free diet. A decrease in FC and FZRP was observed in each infant over a 4-week diet period. During this time, clinical symptoms (intestinal, atopic dermatitis) resolved in all the infants. However, the values obtained were still higher than in the control children, indicating a slower resolution of intestinal allergic inflammation than of allergy symptoms from a few days to 43 weeks. A slightly faster normalisation of the concentration was observed for FZRP; after a month on a milk-free diet, it already affected half of the infants, while calprotectin normalised in only 40% of the subjects. 

This is the first publication describing FZRP concentrations in young infants with MPIAP. FZRP is considered to be the best marker of the increased permeability of the small intestine [30]. The higher concentrations of FZRP in children with MPIAP, in which the lesions are primarily in the colon, indicates that there is also increased small bowel permeability in this disease. On a milk-free diet, allergic inflammation in the colon subsides (stools with blood strands retreat), and the increased permeability of the small intestine decreases.

Sheen et al. found that serum ZRP levels were higher in children with atopic dermatitis than in healthy children [31]. In the study group of children with MPIAP, one-quarter had also atopic dermatitis.

In a group of healthy infants (control group) aged 1–3 months, the median FZRP concentration was 54.1 ng/mL (36.6–101.9 ng/mL). Other values were obtained by Łoniewska et al. when studying 73 infants at 1 month of age 139.61 ng/mL (29.38–712.03 ng/mL). Their study showed that high concentrations of FZRP at 1 month of life will remain at 24 months and, similarly, low concentrations will continue to be low [32].

Niewiem and Grzybowska-Chlebowczyk studied serum levels of ZRP in 49 children aged 7–60 months with IgE-dependent CMPA and 25 with non-IgE-dependent CMPA. They found that the FZRP concentrations were higher in children with non-IgE-CMPA [33]. 

The dysregulation of the zonulin pathway and subsequent “gut leakiness” is caused by increased intestinal permeability and has been associated with the pathogenesis of both intestinal and extraintestinal autoimmune, inflammatory, and neoplastic disorders [30]. High FZRP activity was found in coeliac disease, non-coeliac gluten sensitivity, irritable bowel syndrome, inflammatory bowel disease (Crohn disease, CD and ulcerative colitis, CU), type 1 and 2 diabetes, obesity, and multisystem inflammatory syndrome [34,35,36,37,38,39,40,41,42,43]. FZRP may serve as another biomarker of intestinal damage in inflammatory bowel diseases (IBD), next to FC [38,39,40,41].

FZPR secretion was shown to be regulated by pro-inflammatory cytokines, mainly IL-6 [24,43]. FZPRs are elevated in cigarette smokers [40]. There is no knowledge as to whether passive smoking has an effect on the level of FZPR in infants—especially very young ones. 

The ratio of lactulose to mannitol in urine after oral administration is a recognised test to assess intestinal permeability. It does not correlate with the amount of FZRP in healthy adults, but it does correlate with it in overweight and obese individuals [36].

In our examination, in the control group, the FZRP concentrations were correlated positively with the FC concentrations (r = 0.07, *p* = 0.0026). This relationship was not observed in the study groups (CMPA_o_, CMPA_1_). FZRP was also strongly correlated with FC in CD and CU [38,41].

Calprotectin was the first description, in 1980, by Fagerhol [44]. Its earlier descriptions were calgranulin and “L1 protein”, “MRP-8/14” [45]. Its concentration is stable at room temperature for 4–7 days. It is resistant to enzymatic degradation and stable after freezing, which underlies its clinical utility [46]. Calprotectin is a group of cytosolic proteins that includes calcium- and zinc-binding proteins. It has numerous biological functions, plays a regulatory role in the inflammatory response, and has antibacterial (bacteriostatic) and antifungal effects [47]. 

Calprotectin is localised in the cytoplasm of neutrophils, monocytes, and epithelial cells to protect against invading pathogens (bacterial, fungal, and viral). When it is released into the extracellular environment, it can attract receptors that recognise pathogens, activating innate immune and pro-inflammatory mechanisms. In the inflamed epithelium, calprotectin is released from degranulated neutrophils, forming insoluble antimicrobial barriers known as neutrophil extracellular traps [20,48].

FC is used as a marker of intestinal inflammation, primarily so-called neutrophilic inflammation, in which neutrophils aggregate. The activation and death of these cells release large amounts of calprotectin into the intestinal lumen, which is then excreted in the faeces. FC concentrations in stools rise with increased inflammatory activity in the gut and the number of incoming neutrophils that degranulate [20]. Depending on the site of neutrophil degranulation, large amounts of FC increase in body fluids (serum, urine, saline, synovial fluid, and cerebrospinal fluid) or faeces [20,47,48].

High FC levels are described in Crohn’s disease, ulcerative colitis, cystic fibrosis, bacterial infections, cardiovascular and neurological diseases, autism, gastric cancer, and necrotising enterocolitis (NEC) [20]. In clinical practice, FC is recognised as a validated and recommended biomarker of intestinal inflammation in inflammatory bowel disease (IBD), and it is used especially to control the course of the disease, both in children and adults [20,49,50]. Its usefulness in the diagnosis of microscopic colitis, colagenous colitis, and infectious colitis has not been sufficiently studied [51]. It is also used in the differentiation of functional bowel disorders in patients with the diarrhoeal form of IBD [20].

The concentration of FC is higher in children than in adults. In children, FC decreases with age, especially up to the age of 4 years [20]. After this period, in healthy individuals, it does not exceed 50 ug/g, although healthy children can have FC levels of up to 100 ug/g or even higher, probably because of the increased permeability of the intestinal mucosa and differences in intestinal flora [52]. A very high inter-individual variability in FC concentration is observed in infants up to 12 months of age, making it very difficult to use it to assess conditions occurring at this age [53,54]. In Table 4, we have included studies showing FC concentrations in infants up to 12 months of age [32,52,53,54,55,56,57,58,59,60,61,62,63,64]. The median FC concentration in the 30 children in our control group was lower than in the studies presented here, at 113.25 mg/L (13.9–219.9 mg/L). Similar values were obtained by Łoniewska et al., who also studied a population from the same region [32,61], and by Orivuori (Pasture study), who studied 758 healthy 2-month-old infants living in Austria, Finland, France, Germany, and Switzerland. The FC range was higher in the children living in rural areas than in those in urban areas 76.86 mg/L vs. 308.0 mg/L [65]. The same FC concentrations were also obtained by Finnish researchers in 237 infants aged 3 months: 127 mg/L–212 mg/L [66].

Many researchers have found higher FC concentrations in breastfed infants [67,68,69]. The physiological significance of this phenomenon requires further study [70]. In our study, we did not observe this relationship, probably due to the high proportion of breastfed children (90%). The type of delivery has also been shown to influence FC levels in young infants, but research is inconclusive [59,71].

Young infants with symptoms of stools with blood strands (probably FPIAP) were studied by Baldassarre and Altaee. Baldassarre et al. found that FC concentrations were significantly higher in these infants than in healthy infants (325.89 ± 152.31 mg/L vs. 131.97 ± 37.98 mg/L) and decreased by 50% after 4 weeks of a milk-free diet, although the values were still higher than those in healthy infants [72]. These results are similar to ours. In the group with a diagnosis of MPIAP (MPIAP_0_), the FC concentrations were 382.9 mg/L (103.5–822.8 mg/L). After 4 weeks on a milk-free diet, the FC concentrations significantly decreased, by 45%, to 208.4 mg/L (67.9–484.4 mg/L). Similar results were obtained by other researchers [73,74].

Many researchers have confirmed a decrease in FC levels in CMPA infants treated with a milk-free diet [63,64,73,75,76,77]. Xiong et al. published a review of 13 studies involving 1238 infants and older children [78]. They found that infants with CMPA, especially those with non-IgE CMPA, have elevated FC concentrations, which decrease on a milk-free diet. At the same time, the changes in FC concentration before and after provocation are not statistically significant. The FC concentration is potentially a good biomarker for the diagnosis, monitoring, and prediction of CMPA in children. However, due to the influence of numerous factors, primarily age, feeding method, and the occurrence of different forms of CMPA, this requires further research.

Galip et al. ruled out the usefulness of FC determinations in the diagnosis and monitoring of allergic proctocolitis [79].

The diagnosis of MPIAP on the basis of FC concentrations is subject to considerable error, due to the high individual variability of this indicator. FZRP is a better parameter, but this needs further research, as these are the first determinations in infants with MPIAP. An important limitation is the phenomenon of FZRP not being produced in all the individuals, affecting one in five infants. Individuals bearing the heterozygous Hb2-1 or homozygous Hb2-2 polymorphism are zonulin producers. These individuals, with the homozygous Hp1-1 polymorphism, are unable to produce zonulin. Zonulin is also present in human sera and stool [21]. In our study group of children, zonulin was not produced by 16 children (20.6%), who all belonged to the CMPA group (18.6%). This limits the usefulness of this marker in diagnosis. 

It is necessary to test other faecal markers of intestinal inflammation, such as eosinophil cationic protein, beta defensin, or others.

## 5. Conclusions 

In the present study, we investigated two biomarkers found in the stool, FC and FZRP. FC is a biomarker of intestinal inflammation. FZRP is a new marker of increased intestinal permeability. They have the advantage of being non-invasive, highly sensitive, simple to perform, accessible, and easy to use in infants and children. We found that FC and FZRP concentrations increase in young infants with MPIAP and significantly decrease after 4 weeks of a milk-free diet. These biomarkers are therefore useful to assess the decrease in allergic inflammation activity during dietary treatment. 

## Data Availability

The results of the tests are included in the records of the outpatient clinics where the children were treated.

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
