# Peer review of "Can Faecal Zonulin and Calprotectin Levels Be Used in the Diagnosis and Follow-Up in Infants with Milk Protein-Induced Allergic Proctocolitis?"

_nutrients, 2024, doi:10.3390/nu16172949_

Round 1

Reviewer 1 Report

Comments and Suggestions for Authors

This single-center cohort study demonstrated that fecal calprotectin (FC) and zonulin-related proteins (FZRP) concentrations decreased in infants with MPIAP after 4 weeks of treatment with a non-dairy diet, which was associated with resolution of allergic symptoms, and that the infants with MPIAP group had higher concentrations of FC and FZRP than the control groups after treatment with a non-dairy diet. They argued that FZRP is a better parameter than FC to assess the resolution of colitis in infants with MPIAP. This paper is not novel with respect to FC, but does contain new information on FZRP. This manuscript has many redundant statements throughout and needs to be re-write in a concise manner.

Specific comments

1) The result of the ROC curve analysis should be plotted as a line graph to demonstrate the reliability of the cut-off values of FC and FZRP.

2) Table 3 should include ROC curve AUC, positive predictive values and the negative predictive, sensitivity, and specificity.

3) The first paragraph of the discussion (L230-251) would be fine for a review article, but it is too long for an original article. The background information on FPIAP should be included in the introduction.

4) It is not necessary to repeat in detail in the discussion what is described in the results section. The same explanations are repeated over and over again and become boring. Please delete the repetition of explanations and keep the whole discussion concise.

5) Since the FC values of healthy children reported in previous studies are summarized in Table 5, it is not necessary to give specific values for each report in the text (L332-352). General and characteristics such as age-related changes in FC values of healthy children should be described.

6) Although FC values in infants with CMPA reported in the past studies are described, it is difficult to compare the FC values with those in this study because of different measurement conditions. Thus, it is not necessary to include the FC values of past reported studies in the discussion in detail (L354-394). If you want to include the specific FC values, it is better to summarize them in Table 5. It is sufficient to discuss that the FC values are higher in patients than in normal infants, and that the trend is consistent with the present study in that the values decrease with elimination of cow's milk.

7) The conclusion section is too long, please summarize briefly.

Minor comments

1)Scheme I is deformed and strange, probably because it was converted to PDF. In particular, the relationship between the items framed in orange and those framed in blue is not clear.

Author Response

Thank you for your review. I have made the recommended changes to the text, which are highlighted in blue and red.

  1. Two figures were added (Fig.1 and Fig.2) graphically showing the values of calpoprotectin and zonulin concentrations in the studied groups of children. Marked on them is the value that best differentiates the concentrations in children with allergy disclosure from those in the control group
  2. Table 3 is left, which additionally shows the percentage of children in the study groups whose calprotectin and zonulin concentrations are within the ranges that differentiate healthy children from those with milk allergy and shows statistical significance index.
  3. First paragraph of discussion (L230-251) shortened and moved to introduction (highlighted in blue).
  4. The discussion was shortened, and repetitions of the results were removed.
  5. In the text of the Discussion, in addition to Table 5, only three studies of calprotectin concentrations in healthy infants are mentioned, whose values were similar to our results. The results of FC concentrations included in Tab.5 differ from the values we obtained.
  6. In the text of the Discussion, in order to confirm our results, we discuss several studies of calpoprotectin concentrations in children with milk allergy, in which the authors found higher FC concentrations in children with a disclosure of milk allergy and a decrease after a milk-free diet.
  7. I shortened the conclusion.
  8. I sent the diagram and figures to the Editor independently. The Editor assured me that the diagram and figures would be incorporated into the text without distortion.
  9. Thank you for pointing out the misspelling of the word calprotectin.
  10. Table 1 is in the Materials and Methods section because there in the first paragraph I describe the study group

Reviewer 2 Report

Comments and Suggestions for Authors

It was interesting reading your paper, however, I believe several edits are necessary:

1. I am not sure the the word "calpoprotectin" is correctly written. It should be "calprotectin". 

2.  the table with characteristics of the study group should not be in the materials and methods

3 i would consider adding one or two representative graphics of your most significant findings regarding zonulin and calprotectin.

4.  start the discussion section with a paragraph or two as critical analysis of your most significant findings.

5. table 5 also has to be moved to the results section. 

6. include a paragraph of study limitations and future perspective at the end of the discussion section.

7, add a paragraph of clinical utility of your findings at the bottom of the discussion section (before limitations).

good luck!

Comments on the Quality of English Language

minor edits needed

Author Response

Thank you very much for your review. It allowed me to avoid significant mistakes. When correcting the article, I took into account all the recommendations.

  1. I have corrected the word “calpoprotectin” to “calprotectin” in the text and titles of the cited publications. Thank you very much for noticing this error. I marked the changes made in red.
  2. Table 1. with the characteristics of the study group can be found in the chapter Materials and Methods. In the first paragraph of this chapter, I describe the study group
  3. I added two graphs showing the values of calprotectin and zonulin concentrations in the groups of children studied. On the graphs, I marked the value differentiating the group of infants with MIAP from the group of healthy infants
  4. I rearranged the content in the Discussion chapter. As recommended, I started the discussion with a critical analysis of the main findings.
  5. Table 5 is included in the Discussion section because it includes other authors' findings on fecal calpoprotectin concentrations. Its purpose is to show the large individual variability of fecal calpoprotectin values in healthy infants, which is one of the observations arising from our study.
  6. At the end of the Discussion, I added information describing the limitations of the study and perespectives for the future.

7 At the end of the Discussion section, before describing the limitations of the study, I described the clinical benefits obtained from the results presented.

Thank you

Round 2

Reviewer 1 Report

Comments and Suggestions for Authors

Corrections have been made to the comments, but they are still not appropriate. Additionally, the diagram added is inappropriate.

Specific comments.

1) It is inappropriate to display the results of Figures 1 and 2 as line graphs. When the horizontal axis represents the registered patient number, it is not suitable to connect the data for individual patients in a line graph. If you want to show the change before and after the intervention for each patient, connect the values before and after the intervention with a line segment for each patient. Since the controls are different subjects from the patients, it is inappropriate to overlay them as a line graph with a horizontal axis of the registered patient number.

2) The conclusion of the abstract is too long; the last sentence referring to the limitation depicted at the end of the conclusion should be listed at the end of the result section.

4) In L49, " The most frequent food trigger is cow's milk, rarely hen's egg, soya, 49 wheat and corn [10-12 ]."  This sentence should be deleted due to the lack of semantic connection between the preceding and following sentences.

5) As I pointed out in my previous comment, since the subjects and the values of FZRP and FC reported in each paper are summarized in the form of a table, there is no need to reiterate the reported results in the detailed discussion. The information presented in the table should be organized, and the discussion in lines 385-409 should be about half the current length.

Minor comment

1) p=0.0000 should be p<0.0001

Author Response

-

Reviewer 2 Report

Comments and Suggestions for Authors

The authors have significantly improved the manuscript. 

Comments on the Quality of English Language

Minor edits required. 

Author Response

-